# OpenReview forum: "MoCa: Measuring Human-Language Model Alignment on Causal and Moral Judgment Tasks"
_NeurIPS.cc/2023/Conference — NeurIPS 2023 poster_

### Official Review · Reviewer_HXMa · 2023-06-23

**Soundness:** 3 good
**Presentation:** 2 fair
**Contribution:** 3 good
**Rating:** 7
**Confidence:** 4

**Summary:**

The paper examines the causal and moral judgments made by large language models (LLMs) and their alignment with human intuitions. To do this, the researchers created a dataset of stories from 24 cognitive science papers, annotating each story with factors that influence people's judgments, such as norm violations and the avoidability or inevitability of harm.

The authors find that, on an aggregate level, the alignment between LLMs and human intuition has improved with newer models. However, statistical analyses reveal that LLMs and humans weigh these factors differently when making judgments.


**Strengths:**


**Originality**: The paper presents an interesting approach to evaluating large language models (LLMs) by testing their ability to handle tasks related to causal judgments and moral permissibility. The authors have transcribed stories from various papers and used them to test the LLMs, focusing on several factors that influence people's causal judgments and moral dilemmas. This approach is original and provides a new perspective on the capabilities of LLMs.

**Quality**: The authors have meticulously transcribed stories from a number of papers, ensuring a wide range of scenarios for testing the LLMs. They have also collected responses for each story from a crowd-sourcing platform, ensuring a diverse set of responses for analysis.

**Clarity**: The paper is written in a clear and understandable manner. The authors have explained their methodology and the factors they focused on in a detailed and comprehensible way.

**Significance**: The paper contributes to understanding how LLMs handle complex tasks related to causality and morality. This is an important area of research, given the increasing use of LLMs in various applications. The insights gained from this study could be useful for improving these models in the future. The paper also opens up new avenues for research in this area.


**Weaknesses:**

There are a few areas where it could be improved:

1. **Evaluating with more models**: The paper is a bit skewed towards OpenAI models (GPT3 and beyond) for the evaluation. Including more diverse models could provide a more comprehensive understanding of how different LLMs perform on the tasks. This could also help identify whether the observed behaviors are specific to these models or are more generally applicable to LLMs.

2. **Comparing with human performance**: While the paper does a good job of comparing the performance of different LLMs, it does not provide a clear comparison with human performance. This makes it difficult to assess how close the models are to human-level performance on these tasks. Including a human baseline could provide a more meaningful context for the results.

3. **Analyzing incorrect predictions**: The paper could benefit from a more detailed analysis of the models' incorrect predictions. This could help identify common patterns or biases in the models' errors, which could provide insights for improving the models.

4. **Generalizability of findings**: The paper's findings are based on a specific set of stories and tasks. It's unclear how generalizable these findings are to other tasks or domains. The authors could address this by testing the models on a wider range of tasks or by discussing the limitations of their approach in more detail.


**Questions:**

1. **Evaluating with more models**: The paper primarily focuses on GPT-type models and its variants. Could the authors elaborate on why they chose to focus on these models? Would the inclusion of other language models provide different insights?

2. **Comparing with human performance**: The paper lacks a clear comparison with human performance. Could the authors provide a human baseline for these tasks? This could help in understanding how close the models are to achieving human-level performance.

3. **Analyzing incorrect predictions**:  The paper could benefit from a more detailed analysis of the models' incorrect predictions. Could the authors provide more insights into the common patterns or biases in the models' errors? This could potentially help in improving the models.

4. **Generalizability of findings**: The findings of the paper are based on a specific set of stories and tasks. Could the authors discuss how generalizable these findings are to other tasks or domains?

5. **Interpretability and transparency**: The paper presents an analysis of how LLMs reason about causality and morality. However, it's not clear how these insights can be used to improve the interpretability and transparency of these models. Could the authors provide some thoughts on this?


**Limitations:**

The paper addresses the limitations of the work and potential negative societal impact, although not in a dedicated section. The authors acknowledge that their focus is narrow and only on certain aspects of alignment with humans. They caution that their work should not be used to make sweeping and general statements about AI-human alignment. They also note that their moral permissibility task is not a certification task and should not be used as a flat benchmark to beat.

The authors also discuss the ethical considerations of their work, emphasizing the importance of assessing implicit intuitions underlying commonsense reasoning abilities in large language models (LLMs), especially in cases related to morality. They acknowledge that even if a model is not explicitly given the responsibility to make moral judgments, these judgments can appear across many forms of freely generated text. They also recognize the potential for replicating human biases in LLMs and state that this is something they would want to avoid.

In terms of potential negative societal impact, the authors don't explicitly discuss this. However, they do acknowledge the potential for misuse of LLMs and the ethical considerations that come with their use. They also discuss the importance of transparency and consent in their data collection process.

---

> ### Author Rebuttal · Authors · 2023-08-10
>
> Thank you for your thoughtful questions and comments. We will make sure that we incorporate all the feedback.
>
> > Evaluating with more models: The paper primarily focuses on GPT-type models and its variants. Could the authors elaborate on why they chose to focus on these models? Would the inclusion of other language models provide different insights?
>
> We have also included non-GPT-type models such as RoBERTa-large, Electra-gen-large, and Delphi. As far as we know, GPT-type large language models are being widely adopted by the industry for various applications and use scenarios. There is an urgent need to evaluate the behavior of these LLMs to ensure safety and alignment.
>
> Also, some of the latest language modeling training techniques, such as instruction fine-tuning and RLHF, have been mostly implemented on GPT-type models. For example, Alpaca-7B is instruction fine-tuned, and text-davinci-003 and claude-v1 are RLHF fine-tuned. Even though, on a low level, these models seem to share the same GPT-like architecture, the difference in training methods and how that can cause behavioral shifts in LLMs is what we aim to investigate.
>
> > Comparing with human performance: The paper lacks a clear comparison with human performance. Could the authors provide a human baseline for these tasks? This could help in understanding how close the models are to achieving human-level performance.
>
> Thank you for this suggestion. Due to time constraints, we plan to include a human baseline for the final version of our work.
>
> > Analyzing incorrect predictions: The paper could benefit from a more detailed analysis of the models' incorrect predictions. Could the authors provide more insights into the common patterns or biases in the models' errors? This could potentially help in improving the models.
>
> We did a quick error analysis by annotating 80 examples. We sample 10 examples where 4 models made mistakes (text-curie-001, claude-v1, gpt-3.5-turbo, gpt-4) across 2 tasks (causal/moral). We ask the model why they made the choice to use the original story (prompting for an explanation). Examining the explanation, we conduct two analyses: 1). A quantitative analysis for model hallucination; 2). A preliminary qualitative analysis on what the tendencies model shows.
>
> | Model          | Causal | Moral |
> |----------------|--------|-------|
> | text-curie-001 | 8/10   | 3/10  |
> | claude-v1      | 2/10   | 2/10  |
> | gpt-3.5-turbo  | 2/10   | 0/10  |
> | gpt-4          | 0/10   | 0/10  |
>
> For the quantitative analysis, we first annotate for hallucination. A model hallucinates when they re-state the core story in a way that’s inconsistent with the facts provided in the story. For example, if an action is not performed by character A, and the model thinks it is performed by character A, we count this as a hallucination. Due to time constraints, we can only annotate 80 examples, but we can clearly see for the mistakes the model makes, smaller models tend to hallucinate more.
>
> For the qualitative analysis, we read the model explanations on examples where no hallucination is found, and we conclude that larger models indeed have a different preference compared to humans in these stories. On moral stories, Claude-v1, GPT-3.5-turbo, and GPT-4 all seem to be bound by pre-entered moral principles and choose the same action regardless of the story circumstances (we speculate it to be the influence of AI constitution) [1], while humans tend to consider a few more factors (i.e., nuances in the story). For example, when faced with intricate choices, language models tend to take a passive stance. This is true even if acting could save a greater number, even when a smaller number would face certain peril regardless of the action. However, when self-sacrifice is in question, the language model is inclined to metaphorically 'sacrifice' itself. Our observation is based on a limited set of examples. We hope to include a more comprehensive analysis for the final version of the paper.
>
> > Generalizability of findings: The findings of the paper are based on a specific set of stories and tasks. Could the authors discuss how generalizable these findings are to other tasks or domains?
>
> Thank you for your point. Our study offers a specialized evaluation suite for causal and moral judgment tasks, filling a significant gap left by benchmarks like HELM [2] that cover many tasks but overlook these critical areas. While our focus is specific, our benchmark complements others, aiming to enrich the holistic evaluation of language models across various domains.
>
> > Interpretability and transparency: The paper presents an analysis of how LLMs reason about causality and morality. However, it's not clear how these insights can be used to improve the interpretability and transparency of these models. Could the authors provide some thoughts on this?
>
> Thank you for the question. We provide additional error analyses in the response. By pinpointing where LLMs diverge from human judgments, we hope to lay the groundwork for future research to improve model interpretability. Identifying these discrepancies can guide refinements in LLM training, potentially leading to models that resonate better with human intuitions and are more transparent in their reasoning.
>
> Thank you for the review. Did this answer your questions? We are also happy to answer more questions if they arise.
>
> [1] Bai, Yuntao, et al. "Constitutional ai: Harmlessness from ai feedback." arXiv preprint arXiv:2212.08073 (2022).
>
> [2] Liang, Percy, et al. "Holistic evaluation of language models." arXiv preprint arXiv:2211.09110 (2022).

---

> > ### Comment · Reviewer_HXMa · 2023-08-15
> >
> > Thank you for the rebuttal! The answers to my own questions seem persuasive, and I feel that the new analysis is useful. I continue to support the acceptance of this paper.

---

### Official Review · Reviewer_7W7v · 2023-07-06

**Soundness:** 3 good
**Presentation:** 3 good
**Contribution:** 3 good
**Rating:** 7
**Confidence:** 4

**Summary:**


This paper investigates to what extent LLMs can align with human intuitions when making causal and moral judgments. To do this, they collected a dataset of stories from 24 cognitive science papers and created a causal and moral judgment challenge set. They evaluate different LLMs about their alignment with humans and reveal that the implicit preferences can be different even for LLMs trained with the same technique. They find that increasing model sizes actually impact those models’ aggregate-level alignment differently.


**Strengths:**

With the wide spread of LLMs, to understand the alignment between humans and models is an important topic. In this paper:
- They have provided a dataset to understand the human-model alignment, especially in the causal and moral judgment perspective.
- The resources are from cognitive studies which makes it more reliable than the normal text resources.


**Weaknesses:**

 The paper wants to analyze the alignment between humans and models, however it lacks some description of how they conducted the human study.


**Questions:**

- For the factors in Table 2, are they from existing literature reviews or summarized by the authors?
- For the dataset, are those factor labels annotated by the authors or by original cognitive scientists?
- For the human participants, do you have any criteria to select who can participate in the survey?
- Did you educate the participants about different factors before you conduct the survey?
- It seems only part of Fig.2 is visible on my side. Please check that.

---

> ### Author Rebuttal · Authors · 2023-08-10
>
> Thank you for your thoughtful questions and comments. We will make sure that we incorporate all the feedback.
>
> > The paper wants to analyze the alignment between humans and models, however it lacks some description of how they conducted the human study.
>
> Thank you for bringing this up. We have provided a description of our human subject recruitment and how we conducted our human study in Sec A.14 Crowd Sourced Voting Experiment Design and Interface. We will include a screenshot of our study interface and provide a more detailed description in the final paper.
>
> > For the factors in Table 2, are they from existing literature reviews or summarized by the authors?
>
> All the factors mentioned in Table 2 are taken directly from the papers that conducted the original experiments. A graduate student and a professor in moral/causal psychology were involved in carefully discussing and designing these factors. A thorough literature review was conducted to make sure these factors are comprehensive enough.
>
> > For the dataset, are those factor labels annotated by the authors or by original cognitive scientists?
>
> Factor labels for each story are annotated by two data annotators who have backgrounds in psychology. An annotation guideline is included in the supplementary code zip file. Two annotators reached >0.8 inter-annotator agreement (see Appendix Sec A.7 Annotation Guidelines and Sec A.13 Annotation Agreement Calculation).
>
> > For the human participants, do you have any criteria to select who can participate in the survey?
>
> We only recruit participants with English as their first language. Studies that investigate causal and moral intuitions in cross-cultural contexts have often been conducted with translated stories. We believe this is out of the scope of our current study. This information is included in Appendix Sec A.14 Crowd Sourced Voting Experiment Design and Interface.
>
> > Did you educate the participants about different factors before you conduct the survey?
>
> No. Participants were only asked to read a story and select a binary response. We conduct the experiment as close to the original paper’s experimental setup as possible. We do note that some papers did not fully report their experimental setup or participant selection strategy.
>
> > It seems only part of Fig.2 is visible on my side. Please check that.
>
> We apologize for the inconvenience. It seems that this is an issue caused by the LaTex compilation. A possible solution could be using Adobe Reader to open the PDF file. We will fix this for the final version of our paper.
> Thank you for the review. Did this answer your questions? We are also happy to answer more questions if they arise.

---

> > ### Comment · Reviewer_7W7v · 2023-08-13
> >
> > Thanks! I have read the authors' rebuttal and do not have further questions.

---

### Official Review · Reviewer_duEY · 2023-07-07

**Soundness:** 4 excellent
**Presentation:** 3 good
**Contribution:** 4 excellent
**Rating:** 7
**Confidence:** 2

**Summary:**

This model presents a new challenge set of hard edge cases intended to test models understanding of the nuances of the directness of causation and moral culpability, by collecting them from a set of cognitive science papers. This has the clever effect of not only getting challenging stories, but those which would vary along specific features important to humans.

They test LLMs on those outputs to measure agreement with human intuitions; and annotate those cases for a set of features so that one could draw insight from those disagreements.



**Strengths:**

It is a well-written and well-considered paper which both presents a new useful challenge set, and utilizes it to provide interesting analysis of LLM tendencies in causal culpability and moral judgments.  It could clearly lead to further uses both in the evaluation of new models and in further analysis. The literature review is, as far as I could tell, comprehensive.

The work seem rigorous throughout - I appreciate the thorough explorations with personas and automatic prompt engineering, which alleviate worries about the normal fickleness of prompt choice.

**Weaknesses:**

- The size of the challenge set (around 200 stories I believe) is somewhat limited; I don't think that that's too much of a worry for such a challenge set, so I wouldn't view it as a major weakness.
- quibble: A seemingly left-over note on line 232: " This is very very interesing, make the flow better."

**Questions:**

- I'd be very curious about which personas and prompts would lead to worst-case performance for various models, since that might give insight into how the models go awry.
- The improvements in alignment with human judgements from adopting a utilitarian/consequentialist framing is fascinating. However, that doesn't mean that all humans have a utiliarian framing.  Is there any concern that measuring against the average of human judgements might ignore variance between different humans on such judgement tasks?


**Limitations:**

The ethical considerations section seems thoughtful, and I see no unaddressed limitations.

---

> ### Author Rebuttal · Authors · 2023-08-10
>
> Thank you for your thoughtful questions and comments. We will make sure that we incorporate all the feedback.
>
> > quibble: A seemingly left-over note on line 232: " This is very very interesing, make the flow better."
>
> Thank you for pointing this out – we have removed this in our paper.
>
> > I'd be very curious about which personas and prompts would lead to worst-case performance for various models, since that might give insight into how the models go awry.
> We describe the best and worst persona in Appendix Sec A.4, lines 696-700. We have copied and pasted the section below for ease of reading:
>
> For both models, the persona that most closely aligned with our collected responses for Causal Judgment Task is “Emily White is a Republican who got fed up with the party”, and the least aligned persona is “Angela Campbell is a woman who likes to party until the sun comes up”. For Moral Permissibility Task, the most aligned persona for both models is “Allen Lee is a really cool guy”, and the least aligned persona is “Paul Brown is an anti-vaccine activist”.
>
> We find this very interesting. We have used Prolific (a crowd-sourcing website) to provide labels for our challenge set. This is also a popular website where Psychology studies use to recruit participants. Whether there is a political bias in LLM is a fascinating topic and beyond the scope of our paper, but we would like to investigate it in the future.
>
> > However, that doesn't mean that all humans have a utiliarian framing. Is there any concern that measuring against the average of human judgements might ignore variance between different humans on such judgement tasks?
>
> There is a difference between what prompt aligns LLMs best with humans and the actual responses produced for each story. The prompt that makes LLMs align best with human reasoning is “adopt a utilitarian/consequentialist framing”, but this does not mean that human participants explicitly considered consequentialist principles when providing judgments for the different scenarios. Indeed,  there are scenarios  where human participants make judgments that go against utilitarian principles. For example,  in one of the stories, participants are asked whether it's morally permissible to kill 5 children of a particular race in an orphanage in order to save hundreds of other children in WWII. Here, participants overwhelmingly refuse to kill even if it means that this would have resulted in the greatest number of lives saved. So, as the reviewer suggests, while utilitarian principles certainly seem to play an important part in human moral reasoning, it's not the only consideration that matters to humans.
>
> The reviewer is also right in pointing out that different people's moral intuitions vary substantially. For some individuals, utilitarian principles matter a lot, whereas, for other individuals, deontological rules that prescribe what actions are right or wrong (e.g., do not kill) have a strong impact on their judgments. Currently, our evaluation of LLM alignment is against the aggregate of human judgments, and here it looks like prompting the LLM to adopt a utilitarian framing leads to the best alignment. That said, it's possible that different prompts would help the model to align better with different subgroups of human participants. While we are ultimately interested in better capturing the variance in human moral and causal intuitions, we chose not to do this for the current paper.
>
> Thank you for the review. Did this answer your questions? We are also happy to answer more questions if they arise.

---

> > ### Comment · Reviewer_duEY · 2023-08-13
> >
> > Hi authors! Thank you for the rebuttal; the answers to my own questions seem persuasive, and I feel that the new analysis (in the rebuttal to other reviews) is useful . I continue to support the acceptance of this paper.

---

### Official Review · Reviewer_DYrH · 2023-07-11

**Soundness:** 2 fair
**Presentation:** 3 good
**Contribution:** 2 fair
**Rating:** 6
**Confidence:** 4

**Summary:**

This paper focused on large language models' causal and moral intuitions and investigated the alignment between LLMs and humans' causal and moral judgments. For this purpose, the authors collected story datasets from the field of cognitive science and manually annotated each story with human judgments and underlying latent factors. Based on this dataset, a diverse range of LLMs with different model scales and training methods are evaluated. The authors then statistically revealed that LLMs weigh factors differently than humans, indicating divergent implicit preferences and emphasizing the importance of curated datasets and cognitive science insights in understanding model preferences and alignment.

**Strengths:**

* This paper is well-motivated by philosophy and cognitive science and focused on an exciting topic, LLMs' causal and moral intuitions. Such an interdisciplinary insight would benefit the better understanding of LLMs' behaviours.
* The authors summarized a systematic framework of the underly latent factors of casual and moral judgements based on cognitive science, which might help improve the interpretability of LLMs.
* The constructed judgment dataset is high-quality, with a well-designed annotation protocol and high inter-rater agreement (>0.8).
* The authors benchmarked the alignment level between humans and a wide range of LLMs. They also conducted comprehensive analyses and made inspiring conclusions like those in Sec. 4.2.2, e.g., differences in Benefits.

**Weaknesses:**

* The constructed dataset is too small, and the coverage is limited. Two hundred six instances are highly insufficient to investigate LLMs' properties which might make the conclusion biased. This can be observed in Table 3 (a). The relatively high bootstrapped confidence interval indicates a high variance and unreliable results. This is my biggest concern of this work.

* Some essential results need more in-depth analysis and explanation. (1) The unnatural results in Table 3(a) need more analysis. Why did the aligned and larger Alpaca-7B get lower Agg than GPT3-curie-v1 on Causal Judgement? Why did davinci-002 outperform the well-aligned davinci-003 on moral judgement？ (2) The authors should provide some (even initial) analysis of the differences introduced in Sec. 4.2.2 though they are attractive.

**Questions:**

* How do you explain the unnatural results in Table 3(a): the aligned and larger Alpaca-7B got lower Agg than GPT3-curie-v1 on Causal Judgement; GPT-4 performed even worse than davinci-003 on Causal Judgement; davinci-002 outperformed the well-aligned davinci-003 on moral judgement.
* Would you release your Judgement dataset?

**Limitations:**

The authors have discussed the ethical considerations in Appendix A. However, the authors should also include more discussions of limitations, like the small dataset and variance of the results, as stated above.

---

> ### Author Rebuttal · Authors · 2023-08-10
>
> Thank you for the thoughtful feedback and for appreciating that a carefully curated set of examples and insights from philosophy and cognitive science can help provide a deeper understanding of the implicit biases and tendencies in LLMs. We will make sure to incorporate all the suggestions you have.
>
> > The constructed dataset is too small, and the coverage is limited.
>
> Indeed our dataset size is on the smaller side compared to the typical datasets to evaluate LLMs. However, each of our stories comes from a published, well-sourced paper that investigates a specific part of human intuition on causal or moral reasoning. Each of these stories was carefully designed and, in the original study, was used to get responses from 50-100 human participants. This is drastically different from the typical machine learning / natural language processing, where data were often crowd-sourced and only annotated by 2 or 3 annotators.
>
> Moreover, our collected set of stories represents a wide range of factors (intuitions) that build up the human moral/causal reasoning process. Our coverage on the spectrum of known bias/intuitions is more than sufficient. We believe our work can inspire the community to design and curate more datasets like ours to evaluate LLM behaviors.
>
> > The authors should provide some (even initial) analysis of the differences introduced in Sec. 4.2.2 though they are attractive.
>
> We designed a quick analysis experiment to investigate whether the preference difference is due to actual preference difference or model hallucinations.
>
> We annotated 80 examples by sampling 10 examples where 4 models made mistakes (text-curie-001, claude-v1, gpt-3.5-turbo, gpt-4) across 2 tasks (causal/moral). We ask the model why they made the choice to use the original story (prompting for an explanation). Examining the explanation, we conduct two analyses: 1). A quantitative analysis for model hallucination; 2). A preliminary qualitative analysis on what the tendencies model shows.
>
> |Model|Causal|Moral|
> |-|-|-|
> |text-curie-001|8/10|3/10|
> |claude-v1|2/10|2/10|
> |gpt-3.5-turbo|2/10|0/10|
> |gpt-4|0/10|0/10|
>
> For the quantitative analysis, we first annotate for hallucination. A model hallucinates when they re-state the core story in a way that’s inconsistent with the facts provided in the story. For example, if an action is not performed by character A, and the model thinks it is performed by character A, we count this as a hallucination. Due to time constraints, we can only annotate 80 examples, but we can clearly see for the mistakes the model makes, smaller models tend to hallucinate more.
>
> For the qualitative analysis, we read the model explanations on examples where no hallucination is found, and we conclude that larger models indeed have a different preference compared to humans in these stories. On moral stories, Claude-v1, GPT-3.5-turbo, and GPT-4 all seem to be bound by pre-entered moral principles and choose the same action regardless of the story circumstances (we speculate it to be the influence of AI constitution) [1], while humans tend to consider a few more factors (i.e., nuances in the story). For example, when faced with intricate choices, language models tend to take a passive stance. This is true even if acting could save a greater number, even when a smaller number would face certain peril regardless of the action. However, when self-sacrifice is in question, the language model is inclined to metaphorically 'sacrifice' itself. Our observation is based on a limited set of examples. We hope to include a more comprehensive analysis for the final version of the paper.
>
> Beyond performing some error analyses by asking models to self-explain, we additionally found the following larger trends across models:
>
> 1). Non-monotonicity: the alignment to human biases does not necessarily increase when the model size increases. We speculate that alignment is an area where the inverse scaling law applies [2].
>
> 2). Heterogeneity: interestingly, but perhaps not surprisingly, models that used the same “training method” (such as RLHF) and fine-tuned for human preferences do not have the same implicit biases (see the difference between Claude-v1 and GPT3.5-turbo).
>
> 3). Egocentric Bias vs. Preference of Others: Humans are ego-centric and often make self-beneficial decisions. However, when asked to judge the behaviors of others, we prefer other people to be altruistic (see [3] for examples of human preferences vs. self-choices in scenarios involving self-driving cars). This difference will make models trained on human preferences to be different from human biases. This highlights the importance of a challenge dataset like ours that can measure the implicit bias of the LLMs in order to accurately measure where models and humans align (is it with human preference, or is it with human intuition/bias).
>
> > Some essential results need more in-depth analysis and explanation. Table 3(a): Why did the aligned and larger Alpaca-7B get lower Agg than GPT3-curie-v1 on Causal Judgement? Why did davinci-002 outperform the well-aligned davinci-003 on moral judgement?
>
> Alpaca-7B is fine-tuned on a dataset to be aligned with RLHF-trained text-davinci-003 model. However, it is actually smaller than GPT3-curie-v1, which has 13B parameters. It is unclear exactly why the differences exist – but as discussed above, we believe alignment with human intuition does not follow the traditional scaling law. We also want to point out that alignment with human preference is not the same as aligning with human intuitions.
>
> Thank you for the review. Did this answer your questions? We are also happy to answer more questions if they arise.
>
> [1] Bai, Yuntao, et al. "Constitutional ai: Harmlessness from ai feedback."
>
> [2] McKenzie, Ian R., et al. "Inverse Scaling: When Bigger Isn't Better."
>
> [3] Kallioinen, Noa, et al. "Moral judgements on the actions of self-driving cars and human drivers in dilemma situations from different perspectives."

---

> > ### Comment · Reviewer_DYrH · 2023-08-21
> > **Thanks for response**
> >
> > Thanks for the detailed response. I think most of my concerns have been addressed, and I have raised my score to support the acceptance. Please include your additional experiments (more sampled examples would be better) in your final version, which would make this work more convincing.

---

### Official Review · Reviewer_sFvB · 2023-08-01

**Soundness:** 3 good
**Presentation:** 4 excellent
**Contribution:** 3 good
**Rating:** 7
**Confidence:** 3

**Summary:**

The motivation of this paper is that people constantly make lots of causal and moral judgments to reason about why did what things and why. This paper contributes a dataset of stories compiled from cog sci papers, with detailed annotation of the factors that contributes to the human judgment. Then, the paper looks at how LLMs make judgments, and check the alignment with humans.

**Strengths:**

- The paper addresses an important topic to check the causal and moral reasoning and the alignment of LLMs with humans
- The proposed dataset looks solid and well-annotated
- The analysis provides insights to the community to develop safer and more aligned LLMs.

**Weaknesses:**

- The size of the dataset is a bit limited, 144 causal stories and 62 moral stories, making the insights drawn upon them be not extensive enough
- The yes/no binary answer is reasonable, but analyzing LLMs behavior using a binary classification task might have a little signal-noise ratio. There needs to be lots of human annotation to evaluate the reasoning quality of LLMs, and whether any misalignment or unsafe reasoning was provided apart from the binary answer.

**Questions:**

1. Are there domain experts in moral psychology / philosophy involved in the design process of this paper? How do you make sure the factors in 2a and 2b are comprehensive and can explain for all the judgment decisions? I saw the appendix A.1, but I would like to see one dedicated paragraph for each of Table 2a and 2b, describing the rationale behind each factor and how they correlate with human intuitions in the main text in the next version of the paper.

2. Can the authors let LLMs to output its reasoning, and then annotate what type of tendencies LLMs show in its reasoning (maybe doing it on a subset, e.g., 50 samples)?

[I have read the rebuttal, and acknowledge the author's effort into it. I'm supportive of the acceptance of this paper.]

---

> ### Author Rebuttal · Authors · 2023-08-10
>
> We would like to thank the reviewer for their thoughtful review and the positive feedback that our intention to build a challenge set to evaluate and understand causal and moral reasoning of LLMs and their alignment with humans. We will make sure to incorporate all the suggestions.
>
> > The size of the dataset is a bit limited
>
> > A binary classification task might have a little signal-noise ratio
>
> Thank you for bringing this up! We agree with you that these are the limitations of our proposed dataset. However, each of our stories comes from a published, well-sourced paper that investigates a specific part of human intuition on causal or moral reasoning. Each of these stories was carefully designed and, in the original study, was used to get responses from 50-100 human participants. This is drastically different from the typical machine learning / natural language processing, where data were often crowd-sourced and only annotated by 2 or 3 annotators.
>
> We do hope to conduct future work to move beyond binary tasks and come up with other alignment comparisons/tests between LLMs and human responses.
>
> > Are there domain experts in moral psychology / philosophy involved in the design process of this paper? How do you make sure the factors in 2a and 2b are comprehensive and can explain for all the judgment decisions? I saw the appendix A.1, but I would like to see one dedicated paragraph for each of Table 2a and 2b, describing the rationale behind each factor and how they correlate with human intuitions in the main text in the next version of the paper.
>
> All the factors mentioned in Table 2 are taken directly from the papers that conducted the original experiments. A graduate student and a professor in moral/causal psychology were involved in carefully discussing and designing these factors. A thorough literature review was conducted to make sure these factors are comprehensive enough. We agree that Table A1 only has a truncated description of each factor. However, we do provide a full paragraph to describe each factor in Appendix Sec A.1 and Sec A.2. Please let us know if the descriptions are still unclear and if further clarifications would be helpful.
>
> > Can the authors let LLMs to output its reasoning, and then annotate what type of tendencies LLMs show in its reasoning (maybe doing it on a subset, e.g., 50 samples)?
>
> Thank you for the suggestions. We annotated 80 examples by sampling 10 examples where 4 models made mistakes (text-curie-001, claude-v1, gpt-3.5-turbo, gpt-4) across 2 tasks (causal/moral). We ask the model why they made the choice to use the original story (prompting for an explanation). Examining the explanation, we conduct two analyses: 1). A quantitative analysis for model hallucination; 2). A preliminary qualitative analysis on what the tendencies model shows.
>
> | Model          | Causal | Moral |
> |----------------|--------|-------|
> | text-curie-001 | 8/10   | 3/10  |
> | claude-v1      | 2/10   | 2/10  |
> | gpt-3.5-turbo  | 2/10   | 0/10  |
> | gpt-4          | 0/10   | 0/10  |
>
> For the quantitative analysis, we first annotate for hallucination. A model hallucinates when they re-state the core story in a way that’s inconsistent with the facts provided in the story. For example, if an action is not performed by character A, and the model thinks it is performed by character A, we count this as a hallucination. Due to time constraints, we can only annotate 80 examples, but we can clearly see for the mistakes the model makes, smaller models tend to hallucinate more.
>
> For the qualitative analysis, we read the model explanations on examples where no hallucination is found, and we conclude that larger models indeed have a different preference compared to humans in these stories. On moral stories, Claude-v1, GPT-3.5-turbo, and GPT-4 all seem to be bound by pre-entered moral principles and choose the same action regardless of the story circumstances (we speculate it to be the influence of AI constitution) [1], while humans tend to consider a few more factors (i.e., nuances in the story). For example, when faced with intricate choices, language models tend to take a passive stance. This is true even if acting could save a greater number, even when a smaller number would face certain peril regardless of the action. However, when self-sacrifice is in question, the language model is inclined to metaphorically 'sacrifice' itself. Our observation is based on a limited set of examples. We hope to include a more comprehensive analysis for the final version of the paper.
>
> Thank you for the review. Did this answer your questions? We are also happy to answer more questions if they arise.
>
> [1] Bai, Yuntao, et al. "Constitutional ai: Harmlessness from ai feedback." arXiv preprint arXiv:2212.08073 (2022).

---

> > ### Comment · Reviewer_sFvB · 2023-08-11
> >
> > Thank you to the authors for the rebuttal and additional evaluation. I'd support acceptance for this paper.

---

### Decision · Program_Chairs · 2023-09-21

**Decision:**

Accept (poster)

**Comment:**

There is consensus among reviewers about the quality and significance of this work. I agree with the reviewers. In the words of reviewer DYrH, "This paper is well-motivated by philosophy and cognitive science and focused on an exciting topic, LLMs' causal and moral intuitions. Such an interdisciplinary insight would benefit the better understanding of LLMs' behaviours". The authors should ensure they address the reviewers' questions in the camera ready at least as extensively as they have done here.